# Combined Analysis of Metabolomics and Biochemical Changes Reveals the Nutritional and Functional Characteristics of Red Palm Weevil *Rhynchophus ferrugineus* (Coleoptera: Curculionidae) Larvae at Different Developmental Stages

**DOI:** 10.3390/insects15040294

**Published:** 2024-04-21

**Authors:** Mengran Chen, Jintao Kan, Yufeng Zhang, Jinhao Zhao, Chaojun Lv, Baozhu Zhong, Chaoxu Li, Weiquan Qin

**Affiliations:** 1Key Laboratory of Green Prevention and Control of Tropical Plant Diseases and Pests of Ministry of Education, College of Tropical Agriculture and Forestry, Hainan University, Haikou 570228, China; chenmr0680@163.com; 2Hainan Engineering Center of Coconut Further Processing, Coconut Research Institute of Chinese Academy of Tropical Agricultural Sciences, Wenchang 571339, China; kanjt@foxmail.com (J.K.); jinhao12138@outlook.com (J.Z.); lcj5783@126.com (C.L.); chaoxu998@163.com (C.L.); baozhuz@163.com (B.Z.)

**Keywords:** red palm weevil larvae, untargeted metabolomics, nutrients, antioxidant activity

## Abstract

**Simple Summary:**

Edible insects have emerged as a highly sustainable alternative to animal protein, while the potential and value of many insect metabolites have not been fully explored and exploited. Among them, the red palm weevil (RPW) *Rhynchophus ferrugineus* Olivier (Curculionidae: Coleoptera), especially its larvae, has a long history of consumption and a large scale of breeding, but there is relatively little information available regarding its metabolic and biochemical information at different growth stages, which limits the further development of feeding and processing industries. This study aims to explore the changes in conventional nutrient, mineral compositions, and the characteristics of the metabolomics of RPW larvae at the early, middle, and old developmental stages. Results showed that the red palm weevil larvae are rich in proteins and lipids, with oleic acid and palmitic acid being the main fatty acids, and early larvae possess the highest protein, ash, and total phenolic content. A total of 424 metabolites such as lipids, organic acids, organic heterocyclic compounds and so on were detected from the RPW larvae. The Kyoto Encyclopedia of Genes and Genomes (KEGG) analysis results indicated the ATP-binding cassette (ABC) transporter, the citric acid cycle, aminoacyl-tRNA biosynthesis and the mTOR pathway have significant effects during the three developmental stages. Moreover, early larvae showed better antioxidant activities in vitro compared to that of larvae at the middle and older stages. These provides scientific evidence and valuable information for the resource utilization of red palm weevil larvae.

**Abstract:**

In this study, the changes in the conventional nutrient and mineral compositions as well as the metabolomics characteristics of the red palm weevil (RPW) *Rhynchophus ferrugineus* Olivier (Curculionidae: Coleoptera) larvae at early (EL), middle (ML) and old (OL) developmental stages were investigated. Results showed that the EL and ML had the highest content of protein (53.87 g/100 g dw) and fat (67.95 g/100 g), respectively, and three kinds of RPW larvae were all found to be rich in unsaturated fatty acids (52.17–53.12%), potassium (5707.12–15,865.04 mg/kg) and phosphorus (2123.87–7728.31 mg/kg). In addition, their protein contained 17 amino acids with the largest proportion of glutamate. A total of 424 metabolites mainly including lipids and lipid-like molecules, organic acids and their derivatives, organic heterocycle compounds, alkaloids and their derivatives, etc. were identified in the RPW larvae. There was a significant enrichment in the ABC transport, citrate cycle (TCA cycle), aminoacyl-tRNA biosynthesis, and mTOR signaling pathways as the larvae grow according to the analysis results of the metabolic pathways of differential metabolites. The water extract of EL exhibited relatively higher hydroxyl, 2,2-diphenyl-1-pyrroline hydrochloride (DPPH) and 2,2’-azobis (3-ethylbenzothiazoline-6-sulfonic acid) (ABTS) radical-scavenging ability with the EC_50_ values of 1.12 mg/mL, 11.23 mg/mL, and 2.52 mg/mL, respectively. These results contribute to a better understanding of the compositional changes of the RPW larvae during its life cycle and provide a theoretical grounding for its deep processing and high-value utilization.

## 1. Introduction

Amid the relentless rise in global population and environmental degradation, food scarcity is a significant challenge confronting humanity, and researchers are investigating alternative sustenance sources from microorganisms, insects, and algae alongside traditional animals and plants [1,2,3]. Insects have been selected as a pivotal solution to the impending food crisis because of their numerous benefits such as species diversity, high yield, rapid reproduction, and extensive consumption history [4,5]. According to incomplete statistics, over 1900 species of insects are considered as a potential food source for humans at various stages of their life cycles; some of them like crickets, mealworms, ants, grasshoppers, and flies have a high content of fat, protein, and other nutrients and have been widely consumed for a long time [6,7,8,9]. Red palm weevil (RPW), *Rhynchophorus ferrugineus* Olivier (Curculionidae: Coleoptera), as a common edible insect, is considered an important part of the daily diet of the people of its native southeast Asian countries such as Thailand and Indonesia [10,11]. At the same time, RPW is a major pest of tropical palm crops such as dates, oil palms and coconuts, and it has migrated to the Caribbean, the Middle East, North America, and other regions through the continuous export trade of live palm trees [10,11,12,13,14]. Generally, RPW goes through four development stages of egg, larva, pupa, and adult, of which the larva stage is the most harmful and can cause significant damage to the palm crop by feeding on young stem tips or new shoots [15]. But the RPW larvae is still regarded as a delicacy because it has many advantages over traditional livestock raising, such as its higher feed conversion rate, lack of antibiotics, low land and water requirements, and so on [10,16]. Therefore, they are raised commercially and eaten by the people through frying, baking, making satay, or other methods in southern provinces of Thailand [2,16]. At present, RPW larvae can be successfully cultivated in palm tree trunks covered with bark on top or in indoor plastic containers filled with crushed palm stems and pig feed, and they reach harvest standards within 25 to 30 days after release in specific indoor plastic containers with an average yield per container of about 1 to 2 kg [12,17]. Notably, the economic benefits of larval farming are also considerable due to its production profits per kilogram reaching as high as about $8, which fully demonstrates the economic value and market potential of RPW larvae [18].

In general, harvesting at the right development stage will yield the greatest economic benefits, because the nutrient and functional active substances of insects are constantly changing as they grow [19,20]. For example, cicadas at the nymph stage have a more appropriate proportion of n-3 to n-6 fatty acids and a higher content of polyunsaturated fatty acids compared with that of the late nymphs and adults who are rich in protein [20]. The feed utilization efficiency of mealworms decreases gradually with larvae maturation, which means it may not be economical to insist on harvesting final-stage larvae [21]. However, current research on the nutritional composition changes of RPW mostly focuses on the screening and evaluation of feed formulas, improvement of specific nutritional components such as fatty acids, and differences in composition in different breeding areas except for studying the morphological characteristics, life history and population dynamics, crop harmfulness, monitoring and early warning, and comprehensive prevention and control methods [10,11,16,22,23,24]. For instance, Chinarak et al. (2020) found that the crude fat, protein and chitin contents, mineral element, amino acid, and lipid composition of RPW larvae at the same age collected from three different farms in Yala, Nakhon Si Thammarat, and Phatthalung had significant differences due to the different raising methods and feeding compositions [22]. They further pointed out that the long-chain omega-3 polyunsaturated fatty acids in sago RPW larvae could be improved by supplementation of fish oil in its dietary feed [24]. Metabolomics analysis techniques have been widely used to describe and evaluate the changes in primary and secondary metabolites of organisms such as animals, plants, microorganisms, and insects in recent years [25,26]. But there have been no reports on the changes in the nutrition, function, and metabolites of RPW larvae during their development period. Therefore, the overall objective of this paper is to systematically analyze and evaluate the basic chemical compositions, nutritional value (amino acid, fatty acid, and mineral composition), differential metabolites, and antioxidant activity in vitro of RPW larvae in the early, middle, and older developmental stages in order to provide a basis for enhancing understanding of the physiology and selecting the optimal harvest period.

## 2. Materials and Methods

### 2.1. Samples Preparation

The artificially raised RPW larvae were collected from the scientific research base of the Coconut Research Institute of Chinese Academy of Tropical Agricultural Sciences. They were divided into three groups, named early (EL), middle (ML), and older larvae (OL), with a head circumference below 2.8 mm, ranging from 2.8 to 6.0 mm, and longer than 6.0 mm, respectively [27]. Then, each sample was washed with distilled water and subjected to starvation treatment for 8 h. Afterwards, they were treated with liquid nitrogen for 15 s, which was followed by freeze-drying and crushing to obtain sample powder.

### 2.2. Determination of Chemical Composition

The moisture content of fresh RPW larvae, fat, ash, and protein content of the dried powder of RPW larvae was determined according to AOAC methods [28]. The total phenol content (TPC) of three samples was determined by the Folin–Ciocalteu method with gallic acid (GAE) as a standard [29]. All tests were performed in three replicates.

### 2.3. Determination of Amino Acid Composition

The amino acid composition was analyzed using an L8900 automatic amino acid analyzer (Hitachi, Tokyo, Japan) with a post-column derivatization method using ninhydrin. Briefly, 100 g of RPW larvae dried powder was mixed with 10 mL of hydrochloric acid (6 M) in a tube with a stopper and stewed on ice for 5 min. Then, the tube was filled with nitrogen and sealed to hydrolyze at 110 °C for 22 h. The hydrolysate was fixed to a volume of 25 mL and filtered after cooling to room temperature. Afterwards, 1 mL of the filtrate was evaporated to dryness under the reduced pressure, and the residue was dissolved in 2 mL of distilled water to evaporate again. The residue was dissolved in 1.0 mL citric acid buffer (pH 2.2). Last, the obtained solution was passed through a 0.22 μm filter membrane and transferred into a sample bottle for further determination by the automatic amino acid analyzer. Each amino acid of the sample was identified by comparing their retention times with mixed amino acid standards. Quantitative analysis was performed using the external standard calibration of peak areas against concentrations of authentic standards. 

### 2.4. Determination of Fatty Acid Composition

The fatty acid composition was determined using an Agilent 7890a gas chromatograph (GC) equipped with a DB-225 MS capillary column. The mixture of RPW larvae dried powder and petroleum ether was extracted under reflux in a water bath at 80 °C for 8 h and then concentrated under vacuum at 40 °C to obtain RPW larvae oil. The oil sample dissolved in isooctane was mixed with a methanol solution containing 2% potassium hydroxide, shaken with sodium bisulfate, and the upper solution was transferred to the sample bottle for GC detection. GC analysis conditions included a gasification chamber temperature of 280 °C, a temperature program of initial 50 °C to 200 °C at 5 °C/min, and then to 230 °C at 2 °C/min, holding for 10 min. The carrier gas was helium with a volume of 1 mL/min and a split ratio of 10:1. The injection volume of each sample was 1 μL.

### 2.5. Determination of Mineral Composition

The mineral elements of RPW larvae samples were determined by inductively coupled plasma mass spectrometry (ICP-MS) equipped with a concentric nebulizer and a nickel (Ni) sampling cone according to the previous method with slight modifications [30]. First, 25 mg of RPW larvae powder and 6 mL of nitric acid were mixed and pre-digested for 1 h in a digestion tank. Then, 2 mL of H_2_O_2_ was added for further digestion by a CEM Mars 5 microwave digestion instrument (Matthews, NC, USA) under a commonly used procedure. The cooled digestion solution was diluted to 100 mL for testing. The detail working condition was a radio frequency power of 1600 W, plasma gas flow of 15.0 L/min, a carrier gas flow rate of 1.0 L/min, temperature of the atomization chamber of 2 °C, an oxide index of 0.45%, a double-charge index of 1.01%, a peristaltic pump flow rate of 0.1 revolutions per second, a compensation gas flow rate of 0.8 L/min, and a sampling depth of 8.0 mm. The quality control sample was tested once after every 10 samples or standards are tested; 3% nitric acid solution was tested once to ensure that the error did not exceed ±10%. The content of each element in the sample was calculated using the standard curve of each element.

### 2.6. Determination of Untargeted Metabolomics 

#### 2.6.1. Metabolite Extraction

First, 50 mg sample and 1000 μL tissue extract solution (75% methanol: chloroform, 25% H_2_O) were mixed evenly and ground at 50 Hz for 60 s. Then, the above steps were repeated another two times. The obtained solution was cooled in an ice water bath for 30 min after ultrasonic treatment for 30 min, which was followed by a centrifuge for 10 min at 12,000 rpm and 4 °C. The supernatant was dried under a vacuum concentration condition, and the residue was dissolved by adding 200 μL of 50% acetonitrile solution containing 2-chloro-L-phenylalanine (4 ppm). Finally, the solution was filtered by a 0.22 μm membrane, and the filtrate was collected for further testing. Six biological replicates were performed on each sample, and an equivalent mixture from each group is used as quality control (QC) samples for testing.

#### 2.6.2. LC-MS/MS Analysis

Liquid chromatography conditions: a Thermo Vanquish (Thermo Fisher Scientific, 81 Wyman street, Waltham, MA, USA) ultra-high performance liquid chromatography system was used to separate target compounds using an ACQUITY UPLC^®^ HSS T3 (2.1 mm × 100 mm, 1.8 µm) column (Waters, Milford, MA, USA). Mobile phase A was 0.1% formic acid in water, and mobile phase B was 0.1% formic acid in acetonitrile. The column temperature, flow rate and samples injection volume were 40 °C, 0.30 mL/min and 2 μL, respectively. For LC-ESI (+)-MS analysis, the mobile phases consisted of (B2) 0.1% formic acid in acetonitrile (*v*/*v*) and (A2) 0.1% formic acid in water (*v*/*v*). Separation was conducted under the following gradient: 0~1 min, 8% B2; 1~8 min, 8%~98% B2; 8~10 min, 98% B2; 10~10.1 min, 98%~8% B2; 10.1~12 min, 8% B2. For LC-ESI (-)-MS analysis, the analytes were carried out with (B3) acetonitrile and (A3) ammonium formate (5 mM). Separation was conducted under the following gradient: 0~1 min, 8% B3; 1~8 min, 8%~98% B3; 8~10 min, 98% B3, 10~10.1 min, 98%~8% B3; 10.1~12 min, 8% B3 [31].

Mass spectrum conditions were as follows: the mass spectrometric detection of metabolites was performed on an Orbitrap Exploris 120 (Thermo Fisher Scientific, 81 Wyman street, Waltham, MA, USA) with an ESI ion source. Simultaneous MS1 and MS/MS (full MS-ddMS2 mode, data-dependent MS/MS) acquisition was used. The parameters were as follows: the sheath gas pressure, aux gas flow and capillary temperature were 40 arb, 10 arb and 325 °C, respectively. The spray voltage for ESI (+) and ESI (-) were 3.50 kV and −2.50 kV, respectively. The first level ion full scan was performed with a first resolution of 60,000 FWHM at the range of m/z 100–1000, and the collision energy and resolution of the second level fragmentation by HCD were 30% and 15,000 FWHM, respectively. The first 4 signals among the data-dependent scans per cycle were collected for fragmentation, and unnecessary MS/MS information was removed though an automatic dynamic exclusion [32].

#### 2.6.3. Data Preprocessing 

The raw data were firstly converted to mzXML format using MSConvert in the ProteoWizard software package (v3.0.8789), and the R XCMS (v3.12.0) package was used to process the raw data for peak identification, peak extraction, peak alignment, and integration and to identify substances with spectral databases such as HMDB (http://www.hmdb.ca (accessed on 15 December 2023)) [33], massbank (http://www.massbank.jp/ (accessed on 15 December 2023)) [34], KEGG (https://www.genome.jp/kegg/ (accessed on 15 December 2023)) [35], LipidMaps (http://www.lipidmaps.org (accessed on 15 December 2023)) [36], mzcloud (https://www.mzcloud.org (accessed on 15 December 2023)) [37] and the metabolite database built by Panomix Biomedical Tech Co., Ltd. (Suzhou, China) with the parameter set to ppm < 30 ppm. The R package Ropls was used for principal component analysis (PCA) and orthogonal partial least squares-discriminant analysis (OPLS-DA) of sample data; t-tests and fold change (FC) analysis were used to compare the metabolite quantity differences between two groups; and MetaboAnalyst was used to analyze and compare metabolite differences with the KEGG Mapper tool visualizing metabolites and corresponding pathways.

### 2.7. Antioxidant Activities In Vitro of RPW Larvae Extract 

#### 2.7.1. Preparation of RPW Larvae Extract

The dried powders of the RPW larvae were extracted with ethanol or water at a ratio of 1:40 (m:v, g:mL) under magnetic stirring for 1 h. The mixture solution was centrifuged at 4000 r/min for 10 min, and the supernatant was collected as ethanolic or water extract.

#### 2.7.2. DPPH Radical-Scavenging Activity Assay

The DPPH radical-scavenging activity of RPW larvae extract was determined according to the method of Zhang et al. (2022) with slight modifications [38]. After thoroughly mixing 2 mL of sample solution at different concentrations with 2 mL of DPPH solution, the mixture was placed in the dark for 30 min. Then, the absorbance was measured at the wavelength of 517 nm. The DPPH radical-scavenging rate is calculated as follows:(1)$$DPPH\;Radical-Scavenging\;Rate(\%)=\frac{A_{j}-A_{i}+A_{0}}{A_{j}}\times 100,$$
where A_0_, A_i_, and A_j_ are the absorbance of the sample control group, the sample group, and the control group, respectively.

#### 2.7.3. ABTS Radical-Scavenging Activity Assay

The ABTS radical-scavenging activity of RPW larvae extract was determined by the method of Zhang et al. (2022) with slight modifications [38]. Briefly, 1 mL of sample solution at different concentrations was mixed with 3 mL of ABTS solution and placed at 25 °C to react for 30 min. Then, the absorbance of the reaction solution was measured at 734 nm. The scavenging rate on the ABTS radical is calculated as follows:(2)$$ABTS\;Radical-Scavenging\;Rate(\%)=\frac{A_{j}-A_{i}+A_{0}}{A_{j}}\times 100,$$
where A_0_, A_i_, and A_j_ are the absorbance values of the sample control group, the sample group, and the control group, respectively.

#### 2.7.4. Hydroxyl Radical-Scavenging Activity Assay

The hydroxide radical-scavenging activity of RPW larvae extract was determined referring to the method of Zhang et al. (2022) with slight modifications [38]. In brief, 1 mL of sample solution at different concentrations was mixed with 1 mL of FeSO_4_ solution (6 M), 1 mL of salicylic acid solution (6 M), and 1 mL of H_2_O_2_ solution (6 M) in turn. After standing at room temperature for 30 min, the absorbance was measured at 510 nm. The hydroxyl radical-scavenging rate is calculated as follows:(3)$$Hydroxy\;Scavenging\;Rate(\%)=\frac{A_{j}-A_{i}+A_{0}}{A_{j}}\times 100$$
where A_0_, A_i_, and A_j_ represent the absorbance values of the sample control group, the sample group, and the control group, respectively.

### 2.8. Statistical Analysis

All results were expressed as mean ± standard deviation. The Shapiro–Wilk test was conducted using SPSS 17.0 software to confirm the normality of the data (*p* > 0.05). Then, Levene’s test was used to assess the homogeneity of variances, and when *p* > 0.05, analysis of variance was employed. Based on the results of the analysis of variance, the significance of differences was determined using Tukey’s multiple comparison method with a *p*-value < 0.05 considered statistically significant. Graph was performed using GraphPad Prism 8.

## 3. Results and Analysis

### 3.1. Analysis of Chemical Composition

The appearance of three fresh RPW larvae showed significant differences when they were collected from the cultivated plastic containers. The EL had a creamy white body with a light brown to brown head, the ML had a creamy to yellowish body with a reddish brown to brown head, and the OL had brown to dark brown heads, which illustrated that the deepening of head color and the increase in head circumference were closely related to the growth stage of RPW larvae. And the further test results shown in Table 1 are indeed the same: the appearance and anthropometric properties of RPW larvae at three growth stages showed significant differences (*p* < 0.05). The OL possessed the maximum length, diameter, weight, and head circumference compared to that of the EL and ML. This result can be considered an important basis for us to divide the growth stage by comparing the changes in head circumference.

In general, the chemical composition of insects is related to their category, feeding status, reproductive cycle, different parts of the organism, etc. As for RPW larvae at three growth stages, the chemical composition among them is also significantly different (*p* < 0.05). The EL contained the highest content of moisture, protein, ash, and total phenolic compared with the ML and OL. This may be the reason why RPW larvae prefer to feed on the tender and watery soft tissue in the stems of palm plants: because the OL with a relatively low moisture content begins to transition toward adults to quickly adapt to activities such as flight and migration in the external environment of the tree body. Meanwhile, a higher TPC usually indicates better antioxidant, anti-inflammatory, and other biological activities [39]. So, the EL possess the highest TPC (21.10 mg GAE/g) and may have good stress survival ability when comparing to the ML and OL. The highest fat, ash and protein content was found in the ML (67.95 g/100 g dw), EL (3.59 g/100 g dw) and EL (53.87 g/100 g dw), respectively, which indicated the development of RPW larvae was a process of protein and ash consumption and the continuous accumulation of fat. In addition, the protein content of three RPW larvae was all above 30%, which was comparable to soybeans (35%) and 1.5 times more than beef (20%), which illustrated that the RPW larvae may be a potential protein resource [40]. 

### 3.2. Amino Acid Composition Analysis

The protein in food is usually digested into small peptides and amino acids after entering into the body; its quality always depends on the types and amounts of amino acids present, especially essential amino acids (EAA). As shown in Table 2, all RPW larvae contained 17 amino acids, comprising eight kinds of EAA and nine kinds of nonessential amino acids (NEAA). The Glu (6.86–9.88 g/100 g protein) was the amino acid with the highest content in the protein of RPW larvae. The content of Lys and Asp was also relatively high. Therefore, RPW larvae can be used as a Lys supplement in a cereal-based dietary patterns. Meanwhile, the contents of Cys and Met in the protein of RPW larvae were relatively lower. At the same time, the contents of Met and Cys were the lowest except for the missing Try, and their content gradually increased with the growth of RPW larvae. Similar changes could also be seen in Asp, Thr, Val, and Gly. Overall, the EAA and total amino acids (TAA) in the protein of ML were significantly higher than those of the other two larvae stages (*p* < 0.05). However, their EAA/TAA were all approaching the standards for edible proteins recommended by the WHO/FAO, whose EAA/TAA and NEAA/TAA were 40% and 60%, respectively [41].

### 3.3. Analysis of Fatty Acid Composition

Fatty acids are precursors of lipid synthesis and important substrates for cell energy supply [42,43]. The fatty acid composition of insect oil generally changes continuously with its growth and development. The larvae usually contain more palmitic acid and oleic acid [44], which was confirmed in the fatty acid composition of RPW larvae once again (Table 3). Twelve kinds of fatty acids were identified from RPW larvae oils except for the lack of nervonic acid (C_24:1_) in EL. The content of oleic acid (C_18:1_) was determined to be 45.56%, 46.79 and 45.22% in the EL, ML, and OL, respectively. The content of palmitic acids (C_16:0_) ranged from 39.57% to 43.01%, which was relatively higher as well, and there was a significant difference among three samples (*p* < 0.05). Overall, the content of unsaturated fat fatty acids (UFAs) in three samples was slightly higher than that of saturated fat fatty acids (SFAs), and the monounsaturated fat acids (MUFAs) were dominated by oleic acid (C_18:1_) and palmitoleic acid (C_16:1_). RPW larvae oils may also have hypotensive and anti-inflammatory activity, inhibiting endoplasmic reticulum stress and modulating the insulin signaling pathway and other physiological effects, but further verification is needed [45]. At the same time, the SFA/UFA ratios for the EL, ML, and OL were 1:1.09, 1:1.13, and 1:1.13, respectively, which were all close to 1:1. Similarly, the SFA/UFA ratio of palm oil with good stability and frying resistance was close to 1:1, too. Therefore, it is necessary to explore the processing and application characteristics of RPW larvae oil in the future.

The intake of polyunsaturated fatty acids (PUFA) has been confirmed to be closely related to human health; increasing the intake of n-3 and n-6 fatty acids in the diet can alleviate adult migraine and promote ferroptosis-mediated anticancer effects in the acidic tumor environment [46,47]. The oils or fats whose ratio of n-6/n-3 was between 2.3 and 5 would have a relatively better effect on preventing cardiovascular disease [48]. While the PUFA content of RPW larvae oil was found to be relatively low, only ranging from 0.62 to 1.55% (Table 3). And the ratios of n-6/n-3 in the oil of the EL, ML, and OL were 4.67, 3.19, and 5.50, respectively, which were significantly lower than those of soybean oil (9.83) and olive oil (14.2). But they were all close to the recommended health values (<4) reported by the UK Department of Health, especially for the oil extracted from the ML [49]. 

### 3.4. Minerals Composition Analysis

Minerals are one of the seven essential nutrients that are indispensable for the normal development of the human body, and their content varies according to the life stage of insects [50,51]. It can be seen from Table 4 that the content of potassium (5707.12–15,865.04 mg/kg), phosphorus (2123.87–7728.37 mg/kg), and magnesium (1123.21–2030.59 mg/kg) in RPW larvae was relative higher. The high potassium level in RPW larvae might give them better functional activity in reducing the risk of hypertension and related cardiovascular diseases. Iron, zinc, copper, and three other microminerals were also detected in three stages of RPW larvae. All six microminerals had the highest content in EL (*p* < 0.05). Iron is essential in substrate redox, hormone synthesis, DNA replication, repair and cell cycle control, nitrogen fixation, and preventing damage induced by reactive oxygen species (ROS) [52]. Zinc is a component of more than 300 enzymes and other proteins [53]. The content of iron (12.25–79.35 mg/kg, dw) and zinc (49.64–211.00 mg/kg, dw) in RPW larvae was found to be much higher than that of pork (7.9–8.8 mg/kg, 17.4–19.3 mg/kg) and chicken (3.7–6.8 mg/kg, 6.8–12.9 mg/kg) [54], which showed that the intake of a certain amount of RPW larvae not only provides the essential trace elements for the human body but also helps prevent and treat iron deficiency anemia.

### 3.5. Metabonomic Analysis

#### 3.5.1. Characterization of Metabolites

To better understand the metabolic changes during different growth stages of RPW larvae, untargeted metabolomics analysis was conducted using the UPLC-MS/MS platform. All metabolite peaks were assigned by a secondary mass spectrometry database. Representative base peak chromatograms (BPC) of typical samples were obtained using the validated UPLC-MS method in positive and negative ion modes (Figure 1a,b). In metabolomics studies based on mass spectrometry techniques, quality control (QC) is essential to obtain reliable and high-quality metabolome data. Further analysis revealed that the QC samples exhibited a clustering trend, indicating good reproducibility within a 95% confidence interval (Figure 2a,b).

In the RPW larvae at different growth stages, 424 metabolites were detected. Excluding those that could not be categorized, the 322 metabolites included 87 lipids and lipid-like molecules, 79 organic acids and derivatives, 48 organ heterocyclic compounds, 33 organic oxygen compounds, 30 benzenoids, 15 organic nitrogen compounds, 14 phenylpropanoids and polyketides, 13 nucleosides, nucleotides, and analogues, 2 alkaloids and derivatives, and 1 homogeneous non-metal compound based on their characteristics, which were grouped into 10 classes (Table 5). Lipids and lipid-like molecules account for the highest proportion of 27.02%, which was followed by organic acids and derivatives (24.53%). 

Amino acids. A total of 58 types of amino acids, peptides, and their analogues were identified, and there were 17 types of proteinogenic amino acids that are beneficial to humans and some key non-proteinogenic amino acids (Figure 3a). Notably, the content of L-lysine, L-arginine, and L-glutamic acid is particularly abundant in RPW larvae. Furthermore, comparative analysis reveals that the relative content of amino acids was highest in EL, followed by ML and OL, which indicated that the content of amino acids is positively correlated to the growth process of RPW larvae. In addition, there were two special amino acid derivatives in RPW larvae, namely pyroglutamic acid and ergothioneine. Similar, pyroglutamic acid was also found in *Hermetia illucens* and *T. molitor*, and it had the effect of regulating lipid metabolism and inhibiting hyperglycemia [55,56].

Organic acids. The relative content of organic acids was the highest in EL, followed by OL, with the lowest relative content in ML (Figure 3b). Citric and malic acids were the major organic acids detected in RPW larvae. These natural organic acids have anti-inflammatory, choleretic, antibacterial, and hypoglycemic properties [57]. Additionally, succinic acid which can serve as an effective target for ameliorating insulin resistance and reducing hepatic glycogen output was also detected in RPW larvae [58].

Lipids. A total of 95 lipids were detected in RPW larvae, including four major classes of lipids: fatty acyls, glycerides, pregnenolone lipids, and sterol lipids (Figure 3c). Palmitic acid, myristic acid, erucic acid, deoxy podophyllotoxin, arachidonic acid, docosahexaenoic acid, and α-tocotrienol were the main lipid metabolites detected in RPW larvae. The relative content of lipids in EL was the highest, which was followed by OL and ML. Generally, the last-stage larva possesses a relatively high content of lipids during the whole development cycle due to the large energy requirements for its metamorphosis into adult insects, and this was consistent with our research findings [20]. Meanwhile, Chinarak et al. (2022) found that feeding oil seeds such as sunflower seeds can increase the PUFA/UFA ratio and total content of PUFA of RPW larvae and improve meat nutritional quality [24]. In this paper, 34 kinds of fatty acids and their derivatives were identified from RPW larvae, among which palmitic acid was the most abundant. As for the UFA, the content of erucic acid was the highest, which was followed by arachidonic acid (ARA). ARA together with linoleic acid and linolenic acid were known as the three essential fatty acids for the human body, which can better control blood lipids and cholesterol as well as reduce arteriosclerosis and cardiovascular and cerebrovascular diseases [59]. In addition, RPW larvae also contained some special steroids, including epitestosterone, dihydrotestosterone, and dehydroepiandrosterone (DHEA), and DHEA had outstanding effects on muscle glucose and lipid metabolism [60].

Carbohydrates. Carbohydrates can provide the main energy for organisms to maintain life activities and have special physiological activities as well. And insects usually contain large amounts of glycogen, glucose, fructose, and trehalose [61]. Twenty-two types of carbohydrates and their derivatives were detected, and the total amount decreases with the growth of RPW larvae (Figure 3d). The highest carbohydrate contents in the EL, ML and OL were attributed to l-erythritol, trehalose and trehalose, respectively. Trehalose had important physicochemical properties and is associated with various biological functions such as regulating glucose homeostasis [62].

Vitamins and coenzyme factors. Vitamins are a class of low-molecular-weight organic compounds which are necessary for maintaining physiological functions in the body, but they are usually not synthesized by the human body and obtained through intake [63]. Thirteen types of vitamins and coenzyme factors were detected in RPW larvae (Figure 3e). The relative content in ML is the highest, and the content within RPW larvae exhibited a trend of initial increase followed by a decrease as the RPW larvae grow. As for the water-soluble B vitamins, pantothenic acid has the highest content, which was followed by riboflavin and nicotinic acid. α-tocopherol was the most abundant form of vitamin E isomers in RPW larvae with its highest content in ML. In addition, the gamma-tocopherol content increased as RPW larvae progressed. The higher content of alpha-tocopherol facilitates RPW larvae’s effective participation in antioxidant mechanisms but also enhances the nutritional value of RPW larvae in human nutrition. Vitamin C was not detected in RPW larvae.

The dynamic variation characteristics of amino acids, lipids, organic acids, carbohydrates, and vitamins were not entirely consistent during the growth process of RPW larvae. The relative concentrations of amino acids, organic acids, carbohydrates, and vitamins all decreased as RPW larvae grew, while lipids exhibited a trend of first decreasing and then increasing. This may be attributed to the fact that insects continuously adjust their metabolic states and metabolites during the developmental process in response to changes in nutritional compounds to meet varying energy demands [26]. During the feeding phase, the larvae consume a significant amount of food, where nutrients such as amino acids promote the proliferation and rapid growth of the larvae’s tissues. After reaching a critical body weight, there is a redistribution of carbohydrates, lipids, and amino acids, shifting from protein to lipid catabolism [64]. For instance, recent research has shown that the level of glutamate decreases while the concentration of eicosapentaenoic acid (EPA) rises as *Coenagrion hastulatum* approaches its metamorphosis [65]. 

#### 3.5.2. Multivariate Analysis of Differential Metabolites

Principal component analysis (PCA) is an unsupervised data analysis method that transforms a multi-dimensional variable system into a low-dimensional one, achieving higher accuracy and reflecting the variations between sample groups and within groups [66]. The contribution rates of the first and second principal components (PC1 and PC2) were calculated as 47.0% and 24.7% after the PCA based on UPLC-MS/MS data, respectively (Figure 4a). Their cumulative contribution rate reached 71.47%, indicating that they could represent the main metabolic substances in the three samples. At the same time, the distance between the three samples was far, and there was good polymerization between the six biological replicates of the same sample, indicating that PCA can effectively distinguish the three samples. However, the polymerization effect of OL was significantly worse than that of EL and ML, indicating that the reproducibility of OL was relatively poor. Similar results could be seen from the cluster heatmaps (Figure 4b) where the three RPW larvae were well clustered within the group, and the differences in the relative amounts of each metabolite in the three samples are clearly shown. 

Orthogonal partial least squares discriminant analysis (OPLS-DA) is a supervised discriminant statistical analysis method which can better showcase the differences between groups, but it may suffer from overfitting. And the overfitting of OPLS-DA was often evaluated by establishing a permutation test model. In this model, the original group labels (Y variable) and control groups of the experiment were randomly shuffled and cross-validated many times (200 times was set in this paper) to calculate the X matrix cumulative interpretation rate (R_X_^2^), Y matrix cumulative interpretation rate (R_Y_^2^), and model predictive ability (Q^2^) values. The closer the three values were to 1, the more stable and reliable the model was. It was generally considered that the model was effective when Q^2^ was greater than 0.5, and the model was excellent when Q^2^ was greater than 0.9 [66]. As the results show in Figure 4c, three samples could be clearly distinguished as well, and their polymerization effect was better than that of PCA, which indicated that the OPLS-DA model could effectively eliminate intra-group noise and random errors (Figure 4c). In addition, the R_X_^2^, R_Y_^2^ and Q^2^ values were calculated to be 0.716, 0.998 and 0.993 (Figure 4d), respectively, indicating that the OPLS-DA model was excellent and has no overfitting.

To clarify the metabolic differences among the three RPW larvae, the univariate analysis was conducted by setting the value of the fold change (FC) to be greater than 2 or less than 0.5 and the variable importance projection (VIP) greater than 1 according to the screening criteria. A total of 157 differentially expressed metabolites were found between the ML and EL, while the ML had an increase in 68 metabolites and a decrease in 89 metabolites when comparing with EL (Figure 5a). Among them, butyryl-L-carnitine exhibited the largest FC value (log2FC—6.82), which was followed by prostaglandin F2a (Log2FC—6.36) and 15-deoxy-d-12, 14-PGJ2 (Log2FC—6.19) (Figure 5c). Similarly, there were 117 differentially expressed metabolites between ML and EL. Compared to ML, OL had an increase in 54 metabolites and a decrease in 63 metabolites (Figure 5b). Notably, Imidazol-5-yl-pyruvate exhibited the largest fold change (log2FC—9.77), which was followed by UMP (uridylic acid) (log_2_FC—9.06) and phosphorylcholine (log_2_FC—7.63) (Figure 5d). It was evident that the differential metabolites decreased as the growth of RPW larvae increased, and the metabolic changes were more obvious during the early growth stage.

To further investigate the significant differences in metabolites between the ML vs. EL group and the OL vs. ML group, all identified metabolites were classified into the following categories according to the Kyoto Encyclopedia of Genes and Genomes (KEGG): amino acid-related compounds, carbohydrates, terpenoids, alkaloids, vitamins, and cofactors, lipids, fatty acid-related compounds, and flavonoids. The Venn diagram depicts the shared differential expression of metabolites between the ML vs. EL group and the OL vs. ML group (Figure 6a). There were 59 differential metabolites that were the key players in response to various growth stages of RPW larvae, and their classifications are shown in Figure 6b. These differentially expressed metabolites could be categorized into 14 groups, primarily focusing on amino acids, fatty acyls, organic acids, sterol lipids, flavonoids, and lipids.

#### 3.5.3. KEGG Annotation and Enrichment Analysis of Differential Metabolites

All differential metabolites from the two comparison groups were matched against the KEGG database to obtain information on the pathways in which metabolites were involved. Enrichment analysis was performed on the annotated results to identify pathways with a high concentration of differential metabolites (Figure 7a,b). The KEGG annotation results showed that differential metabolites in the ML vs. EL group and the OL vs. ML group were annotated to 56 and 57 metabolic pathways, respectively. The amino acid metabolic pathway had the highest annotation count, which was followed by the carbohydrate metabolic pathway, the lipid metabolic pathway, and the metabolism of cofactors and vitamins. As shown in Figure 7c, the differential metabolites between the ML vs. EL group were significantly enriched in ABC transporters, which was followed by the citrate cycle (TCA cycle), arginine biosynthesis, aminoacyl-tRNA biosynthesis, and neuroactive ligand–receptor interaction.

Specifically, the ABC transporter pathway was most active in the ML vs. EL group. ABC transporters in the insects’ body were also involved in the absorption, distribution, and excretion of various drugs and endogenous toxins in normal tissues and organs, thereby providing detoxifying and defensive protection for the body [67]. The highest activity of this pathway suggested a significant amount of material interaction between various cells and the internal environment in EL and ML, indicating frequent metabolite transportation and robust biological activity. Among the OL vs. ML group, the mTOR signaling pathway was the most active, suggesting that the larvae consumed a large amount of food during this period. Nutrients such as amino acids in the food stimulate the secretion of insulin-like peptides, activating the mTOR signaling pathway and promoting the proliferation and rapid growth of the larvae’s tissues [68].

In summary, the differential metabolites presented in the three growth stages of RPW larvae were significantly enriched in amino acid metabolism, carbohydrate metabolism, and lipid metabolism pathways. The regulation of amino acid and carbohydrate metabolism in the body may impact cells’ energy metabolism and biosynthetic processes. At the same time, changes in other enriched pathways may be related to lipid metabolism, vitamin metabolism, and the adaptive regulation of the intracellular and extracellular environments. This suggested that the growth stages significantly affected the metabolism and conversion efficiency of amino acids and fatty acids in the body, thereby influencing the nutritional quality of RPW larvae. This provided a theoretical basis for revealing the differences in RPW larvae quality characteristics at the molecular level.

### 3.6. Analysis of Antioxidant Activities In Vitro

Free radical scavengers, also known as antioxidants, can scavenge excess free radicals in the body and alleviate their damage to the organism [69]. Insects may have good antioxidant potential due to their aerobic metabolism and special physiological structure, but there are still few relevant research reports [70,71]. The DPPH, hydroxyl, and ABTS radical-scavenging activities of RPW larvae extracts are shown in Table 6. It could be seen that the radical-scavenging rate of water and ethanol extract of RPW larvae was constantly enhanced with increased sample concentration, showing an apparent dose–effect relationship. The antioxidant activities of the ethanol extract were significantly lower than those of the water extract at the same test concentration.

As for the scavenging ability regarding DPPH radicals, the water and ethanol extract of the EL at the same concentration had the highest activity among the three samples. It was also higher than that of *Tenebrio molitor*, *Protaetia brevitarsis*, and *Apis mellifera* [29]. Correspondingly, the concentration when the sample eliminated 50% of free radicals (EC_50_ value) of the water extract (1.12 mg/mL) and ethanol extract (15.35 mg/mL) of EL was lower than that of the other two samples as well (*p* < 0.05) (Table 6). At the same concentration, the order of scavenging rate of the water extract of RPW larvae on DPPH radicals was EL > OL > ML. Meanwhile, the scavenging activity of the ethanol extract of ML was slightly higher than that of OL at the same concentration. This might be due to more ethanol-soluble polyphenols in the ethanol extract of ML (Table 1).

Like the DPPH radical-scavenging capacity, the order of the hydroxyl radical-scavenging rate of RPW larvae water extracts during the testing concentration ranges of 5–25 mg/mL was EL > OL > ML. The ethanol extract of ML began to have a higher scavenging rate on hydroxyl radical at the sample concentration excess of 10 mg/mL than that of EL and OL. The hydroxyl radical-scavenging activity of RPW larvae water extracts was higher than that of ethanol extracts at the same concentration. This may be because the water extract contained more active substances that can provide more electrons, such as polysaccharides, flavonoids, phenolic acids, and so on, thus quenching hydroxyl radicals.

The results of the ABTS free radical-scavenging activity test showed that the scavenging activity of both ethanol extract and water extract of RPW larvae on ABTS free radical increased significantly with the increase in concentration. Still, the scavenging ability of ethanol extract on ABTS free radicals was relatively weak. The water extract of EL had the strongest scavenging ability on ABTS free radicals. When the concentration of EL water extract was 5 mg/mL, its scavenging rate reached 95.60%, and the EC_50_ of EL water extract for ABTS radical-scavenging activity was 2.52 mg/mL, which was less than its ethanol extract (17.42 mg/mL).

On the whole, the water and ethanol extract of EL had higher antioxidant activities among the three tests models in vitro than that of ML and OL (*p* < 0.05), and this might be attributed to the highest content of ergothioneine in its extract. As a naturally occurring and highly potent antioxidant, ergothioneine could play a significant role in the prevention and treatment of diseases associated with oxidative stress [72]. Considering that all the larvae were used for these analyses and animals are unable to synthesize ergothioneine by themselves, it is possible that ergothioneine is from the insect diet. Thus, EL and OL with good antioxidant activity should be selected as the harvest target of the RPW larvae farming. In addition, the EC_50_ value on DPPH radicals of RPW larvae was basically lower than that of ABTS radicals. The reason may be that DPPH radicals have a strong binding ability with fat-soluble substances, which is more suitable for evaluating fat-soluble substances’ in vitro antioxidant ability [73].

## 4. Conclusions and Discussion

As the global population continues to grow, edible insects with rich nutrition and environmental friendliness have received widespread attention [1,5]. RPW larvae have a relative higher nutritional value and have had a long consumption history in the southeast Asia region [14]. Providing consumers with more extensive information on the nutritional characteristics of RPW larvae is of great significance for improving their acceptance and promoting their application in the food industry [16]. This paper found that the protein content of EL remained at a high level, while its fat content was relatively low. The protein of RPW larvae consists of 17 amino acids, with glutamic acid being particularly prominent, which is consistent with the findings of Chinarak et al. (2020) [22]. RPW larvae mainly contain UFAs like oleic acid and palmitic acid and are rich in mineral elements such as potassium and phosphorus. Although the amino acid content of EL and ML was higher than that of OL, the content of various EAAs more closely meets the WHO/FAO recommended standard. However, the yield of younger larvae does not have a harvest advantage because the advantage of older larvae in unit weight can compensate for their low amino acid content, but it is worth noting that the feed utilization efficiency of insects will gradually decrease with the aging of larvae [21]. To achieve a balance in nutritional components, integrating the final products of different growth stages of RPW larvae has become an effective strategy.

Furthermore, 424 metabolites were detected from RPW larvae, and 215 metabolites mainly including biological molecules such as lipids, amino acids, and carbohydrates were differentially expressed. And the amino acid metabolism pathway was the most abundant based on the KEGG pathway analysis, which was followed by the carbohydrate and lipid metabolism pathways. These indicated that there was a complex metabolic regulatory network during the growth process of RPW larvae, and the development stages could significantly affect the quality and nutritional structure by regulating the metabolism and conversion efficiency of amino acids and fatty acids in the body. Finally, all RPW larvae were shown to have some antioxidant activity in vitro, suggesting that it may be an antioxidant resource worth developing. However, the potential correlation between all levels of metabolites and biological activities still needs further in-depth study, which will help us better understand the nutritional value and bioactive substance basis of RPW larvae and provide a reference for screening components with specific functions.

Finally, the global trade and inter-state movement of edible insects may increase with the growing popularity of edible insects, especially in Western countries. Thus, it should be noted that as a major pest of palm crops, the large-scale breeding of red palm weevils poses certain ecological risks, and various effective protective measures need to be taken to prevent the overflow of artificially raised RPW and cause economic losses [13,16]. For example, selling RPW larvae for food or as food supplements requires legislative action by government and commercial licensing, and many countries have established laws regulating the importation of edible insect materials to control this risk. For instance, live RPW and other edible insects are not allowed to be imported into New Zealand, but they can be traded through a biosecurity pre-treatment requirement involving the boiling, drying and so on [74]. Thus, biosecurity pre-treatment represents a mutually beneficial solution for both traders and consumers. Meanwhile, thorough investigations must be conducted on microbiological aspects and potential allergens to ensure safety and quality when considering RPW larvae in freeze-dried form.

## Figures and Tables

**Figure 1 insects-15-00294-f001:**
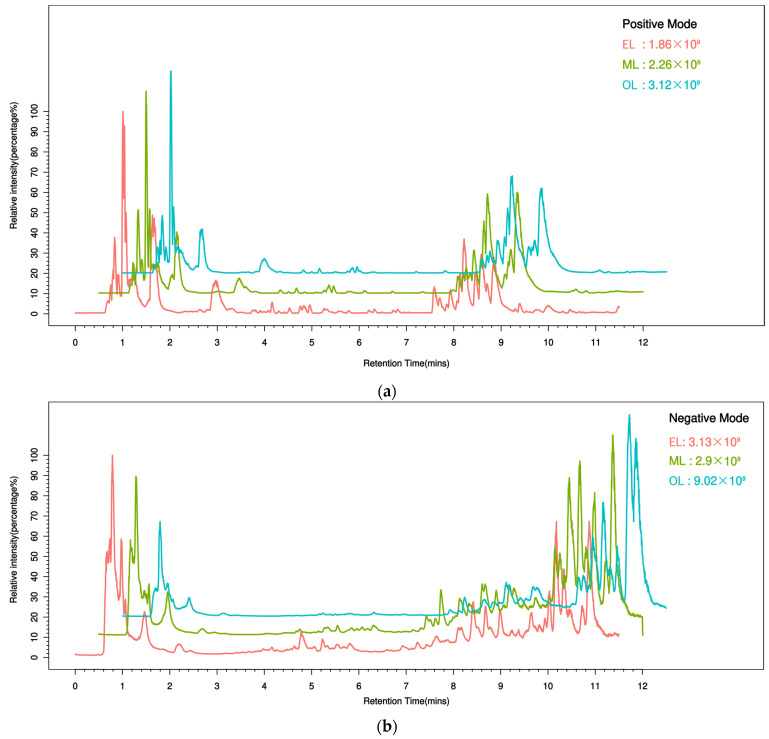
Typical sample base peak chromatogram (BPC) under the mode of positive (**a**) and negative (**b**) ions. (The abscissa represents retention time, the ordinate represents ion intensity, and the top right corner of the plot indicates the maximum ion intensity for each sample. Different colors represent different groups.).

**Figure 2 insects-15-00294-f002:**
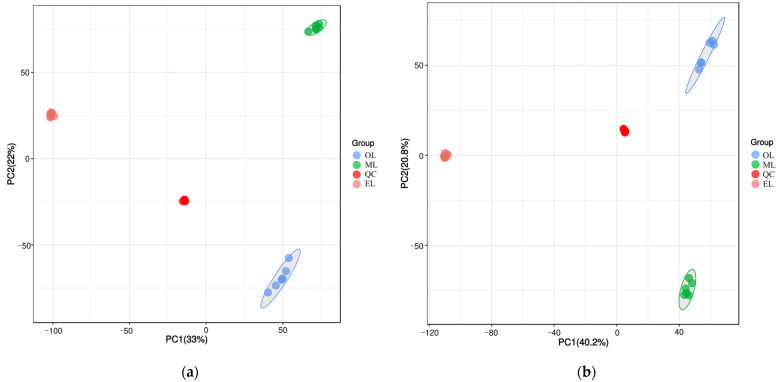
Principal component analysis (PCA) score plot under the mode of positive (**a**) and negative (**b**) ions.

**Figure 3 insects-15-00294-f003:**
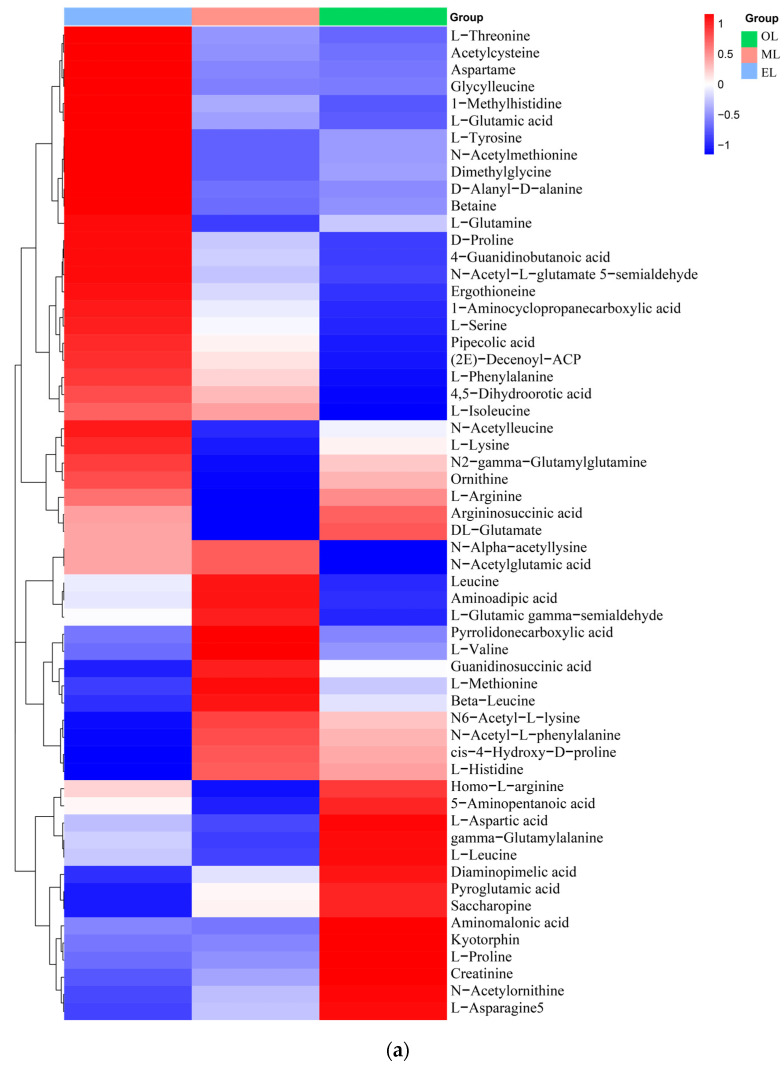
Clustering heatmap of relative contents of (**a**) amino acids, (**b**) organic acids, (**c**) lipids, (**d**) carbohydrates, (**e**) vitamins and coenzyme factors. (A single column or row represents each sample or metabolite. Red squares indicate an increase in the relative number of metabolites, and blue squares indicate a decrease in the relative number of metabolites.)

**Figure 4 insects-15-00294-f004:**
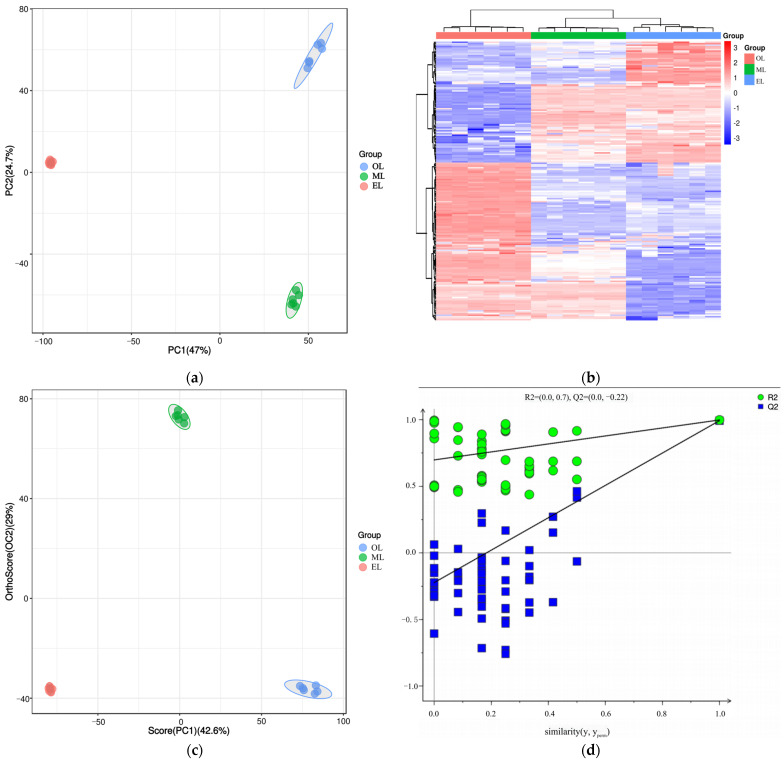
(**a**) PCA score plot, (**b**) cluster heatmap, (**c**) OPLS-DA score plot, and (**d**) OPLS-DA permutation test plot of RPW larvae.

**Figure 5 insects-15-00294-f005:**
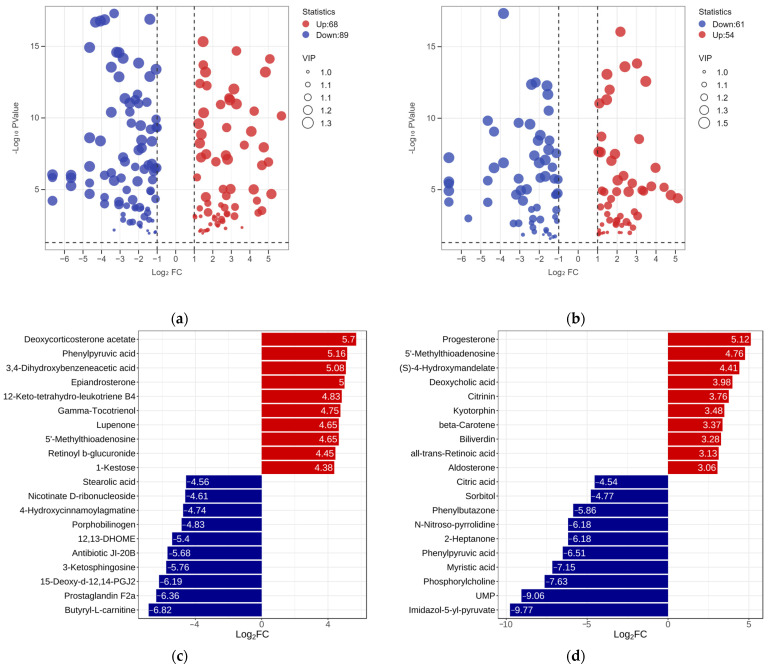
The volcano plot of differential metabolites between the (**a**) ML vs. EL group and (**b**) OL vs. ML group. Bar graph of the difference multiples between the (**c**) ML vs. EL group and (**d**) OL vs. ML group.

**Figure 6 insects-15-00294-f006:**
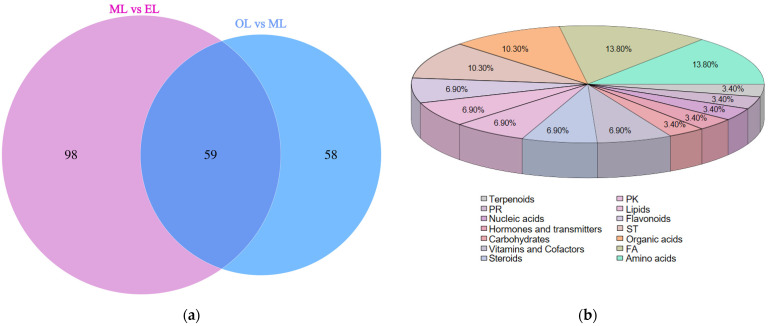
(**a**) Venn diagram of differential metabolites; (**b**) percentage pie chart of metabolite classification.

**Figure 7 insects-15-00294-f007:**
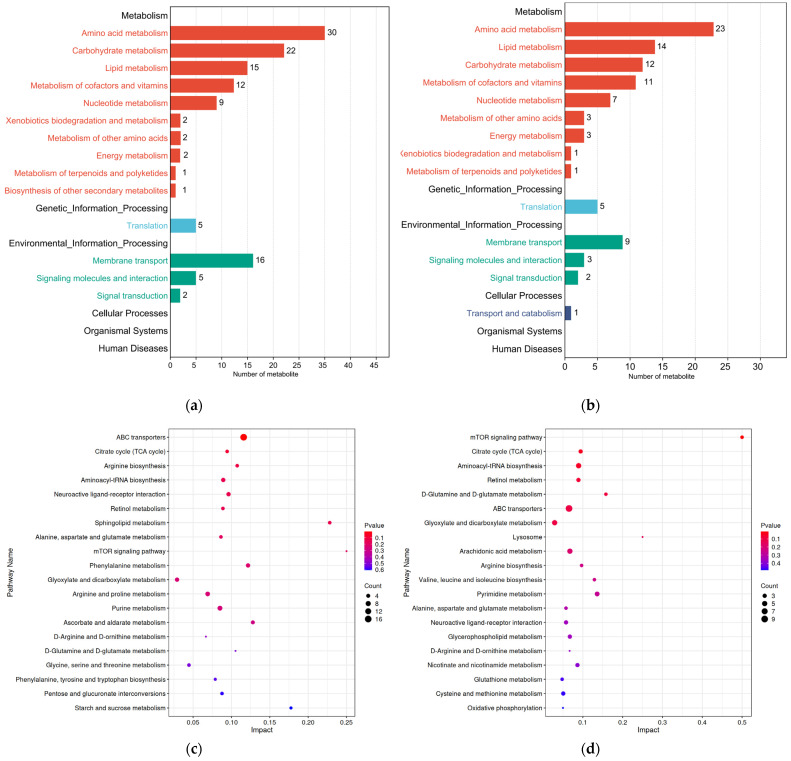
Secondary classification summary of KEGG pathway enrichment results of (**a**) ML vs. El group and (**b**) OL vs. ML group. KEGG pathway enrichment map of differential metabolites between (**c**) ML vs. EL group and (**d**) OL vs. ML group.

**Table 1 insects-15-00294-t001:** Anthropometric parameters and chemical composition of early RPW larvae (EL), middle RPW larvae (ML), and old RPW larvae (OL) ^1^.

	EL	ML	OL
Anthropometric property			
Length (mm)	1.55 ± 0.36 ^c^	3.23 ± 0.35 ^b^	4.99 ± 0.28 ^a^
Weight (g)	0.93 ± 0.22 ^c^	4.60 ± 0.88 ^b^	6.30 ± 0.58 ^a^
Diameter (cm)	0.53 ± 0.13 ^c^	1.33 ± 0.14 ^b^	1.56 ± 0.14 ^a^
Head circumference (cm)	0.17 ± 0.05 ^c^	0.43 ± 0.06 ^b^	0.63 ± 0.02 ^a^
Chemical composition			
Moisture (g/100 g, fw)	80.70 ± 0.34 ^a^	59.95 ± 0.05 ^b^	57.16 ± +0.67 ^c^
Ash (g/100 g, dw)	3.59 ± 0.44 ^a^	2.22 ± 0.09 ^b^	2.17 ± 0.15 ^b^
Protein (g/100 g, dw)	53.87 ± 0.20 ^a^	34.23 ± 0.17 ^b^	32.10 ± 0.21 ^c^
Fat (g/100 g, dw)	18.55 ± 0.18 ^c^	67.95 ± 0.39 ^a^	58.45 ± 0.84 ^b^
Total phenolic content (mg GAE/g, dw)	21.10 ± 0.06 ^a^	6.07 ± 0.16 ^b^	4.47 ± 0.04 ^c^

^1^ The fw and dw represent the abbreviation of fresh weight and dried weight. GAE was gallic acid. Different letters in the right superscript of the same line showed significant differences (*p* < 0.05).

**Table 2 insects-15-00294-t002:** Amino acid composition of early RPW larvae (EL), middle RPW larvae (ML), and old RPW larvae (OL) (g/100 g of protein, dw) ^1^.

	EL	ML	OL
EAA	25.27 ± 1.16 ^b^	26.28 ± 1.17 ^a^	19.46 ± 0.95 ^c^
His	1.72 ± 0.08 ^a^	1.72 ± 0.15 ^a^	1.29 ± 0.09 ^b^
Met	0.68 ± 0.16 ^c^	1.00 ± 0.03 ^a^	0.81 ± 0.04 ^b^
Val	3.76 ± 0.15 ^b^	4.08 ± 0.10 ^a^	3.05 ± 0.24 ^c^
Ile	2.72 ± 0.20 ^b^	3.19 ± 0.22 ^a^	2.44 ± 0.11 ^c^
Leu	4.89 ± 0.13 ^a^	4.88 ± 0.08 ^b^	3.55 ± 0.30 ^c^
Phe	2.77 ± 0.13 ^b^	2.83 ± 0.22 ^a^	2.28 ± 0.09 ^c^
Thr	2.88 ± 0.15 ^b^	3.04 ± 0.06 ^a^	2.29 ± 0.02 ^c^
Lys	5.84 ± 0.17 ^a^	5.54 ± 0.57 ^b^	3.76 ± 0.18 ^c^
NEAA	36.28 ± 0.69 ^b^	40.26 ± 1.33 ^a^	29.18 ± 0.65 ^c^
Cys	0.16 ± 0.05 ^c^	0.46 ± 0.03 ^b^	0.53 ± 0.12 ^a^
Tyr	2.85 ± 0.45 ^b^	3.14 ± 0.36 ^a^	2.14 ± 0.30 ^c^
Asp	5.04 ± 0.70 ^b^	6.53 ± 0.17 ^a^	4.82 ± 0.21 ^c^
Ser	2.84 ± 0.16 ^b^	3.18 ± 0.11 ^a^	2.71 ± 0.13 ^c^
Glu	9.88 ± 0.11 ^a^	9.75 ± 0.37 ^b^	6.86 ± 0.11 ^c^
Gly	3.06 ± 0.3 ^c^	3.62 ± 0.19 ^a^	3.07 ± 0.21 ^b^
Ala	4.39 ± 0.06 ^b^	4.42 ± 0.28 ^a^	3.26 ± 0.11 ^c^
Arg	4.19 ± 0.23 ^a^	4.14 ± 0.05 ^b^	3.10 ± 0.20 ^c^
Pro	3.86 ± 0.12 ^b^	5.01 ± 0.40 ^a^	2.68 ± 0.34 ^c^
TAA	61.55 ± 1.56 ^b^	66.54 ± 2.44 ^a^	48.64 ± 1.60 ^c^
EAA/TAA (%)	41.05 ± 0.94 ^a^	39.49 ± 0.46 ^c^	40.00 ± 0.65 ^b^
EAA/NEAA (%)	69.67 ± 2.73 ^a^	65.27 ± 1.25 ^c^	66.69 ± 1.79 ^b^

^1^ EAA, NEAA, and TAA represent the abbreviation of essential amino acids, nonessential amino acids, and total amino acids. Values are given as mean ± standard deviation from triplicate determinations. Different letters in the right superscript of the same line showed significant differences (*p* < 0.05).

**Table 3 insects-15-00294-t003:** Fatty acid composition of the early RPW larvae (EL), middle RPW larvae (ML), and old RPW larvae (OL) (%) ^1^.

Fatty Acids	EL	ML	OL
Caprylic acid (C_8:0_)	0.11 ± 0.00 ^b^	0.15 ± 0.02 ^b^	0.37 ± 0.12 ^a^
Capric acid (C_10:0_)	0.10 ± 0.00 ^b^	0.13 ± 0.02 ^b^	0.32 ± 0.09 ^a^
Lauric acid (C_12:0_)	0.86 ± 0.01 ^b^	1.09 ± 0.18 ^b^	2.65 ± 0.76 ^a^
Myristic acid (C_14:0_)	1.65 ± 0.01 ^b^	1.69 ± 0.06 ^b^	2.21 ± 0.28 ^a^
Palmitic acid (C_16:0_)	43.01 ± 0.07 ^a^	41.41 ± 0.63 ^b^	39.57 ± 0.41 ^c^
Palmitoleic acid (C_16:1_)	6.01 ± 0.03 ^a^	5.53 ± 0.11 ^c^	5.82 ± 0.09 ^b^
Stearic acid (C_18:0_)	1.80 ± 0.03 ^b^	2.10 ± 0.01 ^a^	1.68 ± 0.05 ^c^
Oleic acid (C_18:1_)	45.56 ± 0.10 ^b^	46.79 ± 0.39 ^a^	45.22 ± 0.81 ^b^
Linoleic acid (C_18:2 n6_)	0.51 ± 0.03 ^b^	0.54 ± 0.01 ^b^	1.31 ± 0.02 ^a^
α-Linolenic acid (C_18:3 n3_)	0.11 ± 0.01 ^c^	0.17 ± 0.00 ^b^	0.24 ± 0.00 ^a^
Behenic acid (C_22:0_)	0.30 ± 0.00 ^a^	0.31 ± 0.00 ^a^	0.2 ± 0.03 ^b^
Nervonic acid (C_24:1_)	ND	0.1 ± 0.00 ^b^	0.42 ± 0.02 ^a^
SFA	47.83 ± 0.01 ^a^	46.88 ± 0.00 ^b^	46.99 ± 0.14 ^b^
UFA	52.17 ± 0.03 ^c^	53.12 ± 0.01 ^a^	53.01 ± 0.63 ^b^
MUFA	51.56 ± 0.76 ^b^	52.42 ± 0.01 ^a^	51.46 ± 0.81 ^c^
PUFA	0.62 ± 0.00 ^c^	0.71 ± 0.00 ^b^	1.55 ± 0.00 ^a^
PUFA/SFA	0.01 ± 0.00 ^b^	0.01 ± 0.00 ^b^	0.03 ± 0.00 ^a^
n-6/n-3	4.67 ± 0.25 ^b^	3.19 ± 0.05 ^c^	5.50 ± 0.05 ^a^

^1^ SFA, MUFA, and PUFA represent the abbreviation of saturated fatty acids, monounsaturated fatty acids, and polyunsaturated fatty acids. Values are given as mean ± standard deviation from triplicate determinations. Different letters in the right superscript of the same line showed significant differences (*p* < 0.05). ND means not detected.

**Table 4 insects-15-00294-t004:** Mineral elements in early RPW larvae (EL), middle RPW larvae (ML), and old RPW larvae (OL) (mg/kg, dw) ^1^.

Mineral	EL	ML	OL
Macrominerals			
Potassium	15,865.04 ± 76.76 ^a^	10,568.25 ± 117.74 ^b^	5707.12 ± 33.35 ^c^
Phosphorus	7728.37 ± 91.87 ^a^	4166.53 ± 41.22 ^b^	2123.87 ± 12.51 ^c^
Magnesium	2030.59 ± 15.54 ^a^	1942.75 ± 46.37 ^b^	1123.21 ± 1.32 ^c^
Sodium	1907.21 ± 9.7 ^a^	1798.43 ± 41.64 ^b^	707.79 ± 3.25 ^c^
Calcium	700.73 ± 22.6 ^a^	594.84 ± 16.74 ^b^	341.86 ± 4.97 ^c^
Microminerals			
Zinc	211.00 ± 4.87 ^a^	112.29 ± 4.56 ^b^	49.64 ± 1.11 ^c^
Iron	79.35 ± 3.23 ^a^	29.48 ± 0.86 ^b^	12.25 ± 0.67 ^c^
Copper	13.91 ± 0.3 ^a^	5.84 ± 0.01 ^b^	5.87 ± 0.04 ^b^
Manganese	10.47 ± 0.22 ^a^	7.16 ± 0.26 ^b^	4.34 ± 0.03 ^c^
Chromium	1.44 ± 0.09 ^a^	0.28 ± 0.02 ^b^	0.16 ± 0.03 ^c^
Nickel	1.18 ± 0.01 ^a^	0.67 ± 0.04 ^b^	0.08 ± 0.01 ^c^
Arsenic	0.07 ± 0.00 ^a^	0.05 ± 0.00 ^a^	0.01 ± 0.00 ^b^
Cadmium	0.33 ± 0.01 ^a^	0.09 ± 0.00 ^b^	0.02 ± 0.00 ^c^
Lead	0.16 ± 0.01 ^a^	0.14 ± 0.00 ^a^	0.01 ± 0.00 ^b^

^1^ Values are given as mean ± standard deviation from triplicate determinations. Different letters in the right superscript of the same line showed significant differences (*p* < 0.05).

**Table 5 insects-15-00294-t005:** Classification and proportion of metabolites in RPW larvae.

Name	Number	Proportion%
Lipids and lipid-like molecules	87	27.02
Organic acids and derivatives	79	24.53
Organoheterocyclic compounds	48	14.91
Organic oxygen compounds	33	10.25
Benzenoids	30	9.32
Organic nitrogen compounds	15	4.66
Phenylpropanoids and polyketides	14	4.35
Nucleosides, nucleotides, and analogues	13	4.04
Alkaloids and derivatives	2	0.62
Homogeneous non-metal compounds	1	0.31
Total	322	100

**Table 6 insects-15-00294-t006:** Antioxidant activities of early RPW larvae (EL), middle RPW larvae (ML), and old RPW larvae (OL) (mg/mL) ^1^.

	Radicals	Samples	EL	ML	OL
EC_50_ value	DPPH	Water extract	1.12 ± 0.01 ^c^	3.94 ± 0.02 ^a^	1.88 ± 0.01 ^b^
		Ethanol extract	15.35 ± 0.03 ^c^	25.83 ± 0.17 ^b^	27.69 ± 0.01 ^a^
	Hydroxyl	Water extract	11.23 ± 0.01 ^c^	16.02 ± 0.06 ^b^	17.73 ± 0.04 ^a^
		Ethanol extract	58.12 ± 0.03 ^a^	36.47 ± 0.08 ^c^	39.43 ± 0.12 ^b^
	ABTS	Water extract	2.52 ± 0.17 ^c^	6.22 ± 0.70 ^a^	4.86 ± 0.35 ^b^
		Ethanol extract	17.42 ± 0.29 ^c^	49.63 ± 0.81 ^a^	45.13 ± 0.26 ^b^

^1^ Values are given as mean ± standard deviation from triplicate determinations. Different letters in the right superscript of the same line showed significant differences (*p* < 0.05).

## Data Availability

All data presented in this study are available on request from the corresponding author. The data are not uploaded in publicly accessible databases.

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
