# Peer review of "Combined Analysis of Metabolomics and Biochemical Changes Reveals the Nutritional and Functional Characteristics of Red Palm Weevil Rhynchophus ferrugineus (Coleoptera: Curculionidae) Larvae at Different Developmental Stages"

_insects, 2024, doi:10.3390/insects15040294_

Round 1
Reviewer 1 Report
Comments and Suggestions for Authors
In this manuscript Chen and co-authors present extensive biochemical composition characterization of a possible insect derived food source from the red palm weevil. Full characterization is vital both to document the levels of nutrients in red palm weevil extracts, but also to identify any antinutrients or unhealthy components. Also studies such as these are necessary to justify the more expensive insect derived material substitution for plant based foodstuffs of known composition and long history of safe human and animal consumption. One need I see lacking in this manuscript is the provision of some supporting evidence that red palm weevils can be mass produced in sufficient quantity to justify this exceptional level of research effort. Perhaps this was mentioned in one of the references? Three larval life stages were analyzed separately demonstrating that there are wide differences in composition of many compounds. The results suggest that mixing various life stages for the final product would be one approach to balancing nutrient content.
Overall the manuscript is of high quality. Detailed determinations were made of the macronutrient, total amino acid, fatty acid, mineral and micronutrient (sans Selenium), and antioxidant content of red palm weevil extracts. The authors employed metabolomic methods to further characterize the broad array of biochemicals present in red palm weevil. This type of data is especially important in demonstrating the differences between different life stages (Fig. 2), enabling pathway analyses with data packed heat maps and PCA plots (Figs. 3, 4, 5, 6, 7). If the input samples are appropriately staged this data could also be used to study the developmental processes undergone during maturation. This experimental work was done in accordance with accepted methods, analyzed and presented appropriately.
Starting with Figure 1, the images must be presented at a higher resolution or as larger images so that the text can be more easily read. This applies to all figures.
Figure 4 legend appears to have text left over from the MDPI provided template ("This is a figure. Schemes follow another format. If there are multiple panels................").
Of some interest, the mention of high levels of ergothioneine in RPWL extract might be discussed further in the sections pertaining to levels of antioxidant protectants. Given that whole larvae were used for these analyses, the ergothioneine, and perhaps other components could be of the insect diet origin.
Author Response
- Supporting evidence for mass production of red palm weevils
Response: We appreciate your suggestion regarding the need for evidence supporting the mass production of red palm weevils. In response, we extend the introduction in the manuscript. We have made a more comprehensive review of the successful cases of large-scale production of red palm weevil in other regions. This includes a detailed description of the methods used, the production costs involved, and the final selling price (lines 59-78).
- Resolution of Figure 1
Response: We acknowledge the importance of clear and readable figures. As suggested, we have increased the size and resolution of Figure 1 and all other graphics in the manuscript to ensure that the text is easy to read.
- Figure 4 legend
Response: We apologize for the oversight in the Figure 4 legend. The text leftover from the template has been removed, and the legend has been revised for clarity.
- Discussion on ergothioneine levels
Response: We agree that the high levels of ergothioneine in red palm weevil (RPW)larvae extract warrant further discussion. We have expanded the discussion in the antioxidant protectants section (lines 600-604) to elaborate on the potential dietary origin of ergothioneine and its implications for the nutritional value of RPWL.
In addition, some other errors in the original manuscript, such as the lack of subscripts for H2O2 in line 163, inconsistencies in title capitalization, and the significant figures in text and tables, etc, were corrected after careful reading and checking of the full text for many times. But there must be some other mistakes in the revised manuscript. Please point out, we will revise it as soon as possible. At last, we would like to express our deep gratitude to you and editors once again. And looking forward to your positive response.
Reviewer 2 Report
Comments and Suggestions for Authors
General Comments:
The introduction and discussion sections lack clarity regarding the manuscript's relation to other cited works and the concerns raised by fellow researchers regarding the growing and harvesting of Rhynchophorus palm weevils for food consumption. A concise explanation of the research's purpose, significance, and alignment with other studies is warranted. The discussion section requires more substantial concluding remarks, especially concerning the biosecurity questions associated with palm weevils worldwide, given their important pest status. While the abstract piqued my interest, extracting key messages beneficial to the practicality of growing and harvesting palm weevil larvae for human consumption proved challenging due to the manuscript's limited connection with other published work on palm weevils and its need for improved structural organization. Though the gathered data holds value, the paper would greatly benefit from an extensive literature review and a more robust connection with relevant reports on the subject. A significant overhaul in the next draft is recommended to improve the manuscript's chances of publication.
Specific Comments: My additional comments encompass some specific observations. Firstly, the paper already employs numerous acronyms related to the methods used. Introducing additional acronyms for red palm weevil larvae and each larval stage may complicate understanding. Secondly, I've noted that on Line 120, "red plan weevil larvae" should be corrected to "red palm weevil larvae." Additionally, it is suggested to specify "older RPWL" instead of "older PRW larvae" to maintain consistency, considering the RPWL acronym is defined earlier in the text. I discourage the use of acronyms for larvae and larval stages, recommending consistent usage of RPW for red palm weevils, RPW larvae, older RPW larvae, etc. This adjustment aims to enhance clarity, maintain a smooth flow throughout the paper, and align with other published works.
Moreover, Sections 91 through 102 are challenging to follow. Firstly, correct the Latin name for red palm weevils to Rhynchophorus ferrugineus instead of Rhynchophorus ferruginous. Include authorship details immediately after each Latin name mention, specifying the order and family, and potentially the species' subfamily or tribe. Eliminate the inaccurate statement that these weevils belong to "the family of Coleoptera weevil"; instead, add the specific family, as Coleoptera is an order. The current representation is unsuitable for a scientific publication, necessitating corrections for accuracy and adherence to scientific standards.
The systematics, pestiferousness, and bioecology of all Rhynchophorus species are thoroughly documented in the most recent annual reviews paper (https://doi.org/10.1146/annurev-ento-013023-121139). I strongly recommend that the authors consult this paper to revise this section, ensuring precision and citing this relevant work, along with other pertinent sources.
Expanding on this, within the same section (91 through 102), the bioecology of Rhynchophorus larvae, which are reared and used for human consumption, has been poorly described and is not easily understandable. Where do these larvae reside inside palms? How are people rearing them and harvesting for food? Rhynchophorus species are destructive pests of palms worldwide, but this aspect has not been adequately explained in this paper. How does it relate to the biosecurity of their movement for food consumption? For instance, the intentional smuggling of Rhynchophorus spp. into a new area for the establishment of populations that could be harvested for human consumption has been proposed, as observed in the introduction of Rhynchophorus vulneratus in Southern California, possibly from Indonesia where they are either harvested from the wild or commercially farmed (see https://doi.org/10.1007/s10340-018-1044-3). Rhynchophorus larvae and pupae are considered delicacies in numerous tropical countries worldwide in Africa, Asia, and South America (see J Ethnobiol 29:113–128; Principes 37:42–47; and EcolFood Nutr 40:13–32), and it is recommended to establish more connections to these other works, especially in the discussion section, which is currently extremely poorly structured and very short in my opinion.
I am eager to provide additional assistance during the review of the next draft of this paper; thank you. Good luck!
Author Response
- Clarity in Introduction and Discussion
Response:We revised the introduction (line 49-106) and discussion (line 616-659) to explain the purpose, significance and consistency of this study with other studies on red palm weevil more clearly. We also expanded the scope of discussion to include more substantive concluding observations. For example, considering their status as important pests, the biosafety pretreatment of red palm weevil by boiling and drying can effectively reduce the biosafety impact on food consumption and the potential risks associated with the introduction of non-native species.
- Acronyms and Consistency
Response: We have carefully reviewed the use of acronyms throughout the manuscript. To simplify understanding, we have minimized the introduction of new acronyms and maintained consistent usage of "RPW" for red palm weevils and "RPW larvae" for their larvae. The suggested corrections have been made on line 111 and throughout the manuscript to enhance clarity. Early larvae is now abbreviated as EL, Middle larvae is now abbreviated as ML and Old larvae is now abbreviated as OL.
- Corrections in Systematics and Bioecology
Response: We have corrected the Latin name of Red palm weevil (RPW) to Rhynchophorus ferrugineus Olivier (Curculionidae: Coleoptera). The inaccurate statement regarding the family of Coleoptera weevil has been rectified, and the specific family has been added (lines 59-62).
- Bioecology and Biosecurity
Response: We have revised the section on the bioecology of Rhynchophorus larvae to provide a clearer and more detailed description of their habitat, rearing, and harvesting for food consumption (lines 59-78). In addition, in the discussion section, we added the discussion on the ecological risks brought by large-scale breeding RPW, and emphasized the necessity of adopting biosafety pretreatment measures (such as boiling, drying, etc.) to effectively prevent the risk of population establishment caused by intentional smuggling of live RPW (line646-659).
- Connections to Other Works
Response: We have established stronger connections to other works on the consumption of Rhynchophorus larvae and pupae as delicacies in various tropical countries. This includes additional citations and a more comprehensive discussion of the cultural and ecological significance of palm weevil consumption (lines 59-78).
In addition, some other errors in the original manuscript, such as the lack of subscripts for H2O2 in line 163, inconsistencies in title capitalization, and the significant figures in text and tables, etc, were corrected after careful reading and checking of the full text for many times. But there must be some other mistakes in the revised manuscript. Please point out, we will revise it as soon as possible. At last, we would like to express our deep gratitude to you and editors once again. And looking forward to your positive response.
Reviewer 3 Report
Comments and Suggestions for Authors
The manuscript entitled "Combined Analysis of Metabolomics and biochemical changes in Red Palm Weevil Larvae on Different Developmental Stages Reveals the Variation of Nutritional and Functional Characteristics" is decent piece of work. However, I recommend some revision to improve the overall quality of the work.
1) During the text it is not entirely clear whether results are expressed based on dry or wat matter. Please check throughout the whole manuscript as it can be a considerable difference.
2) Some abbreviations are not written in full before the first use. This unnecessarily complicates the text for the reader.
3) Please improve the captions of the figures and tables. A figure and table should be able to stand alone, however, this is not completely the case. Further also improve the quality of the figures. Most of them are, for instance, too small to read properly.
4) Check all citations in the text. Some are wrongly presented such as in line 70.
5) Be consistent in the usage of a space between a number and a unit. Check the consistency in the whole manuscript.
6) Check all headings of the manuscript. There are two different ways of presenting used. You have to choose to provide all headings with or without capitals, please revise
7) Check all chemical formulas. Some are wrongly presented such as in line 163 (H2O2).
8) Explain the difference between sample control group and the control group better. This is not completely clear now.
9) Please elaborate on the statistical data analysis. You state that an ANOVA is used, without checking the conditions of normality and homoscedasticity. Please have a look at it and make appropriate changes.
10) Check and correct all significant digits in tables and the text.
11) I think there is made a mistake in the caption of figure 1 (two times positive ion mode) and figure 4 (it seems that the instruction of the template or still there).
12) Is there any evidence why the relative concentrations of AA, OA, carbohydrates and vitamins are not consistent? This is not completely clear, please elaborate on this phenomenon (Lines 457-461).
13) In lines 476-485 the OPLS-DA is discussed, but it is not completely clear for non-experts. Please elaborate on this to make it more clear for the reader.
Comments on the Quality of English Language
In general, the manuscript is well written. Nevertheless, now and then some errors were found which require revision.
1) Check the whole text for grammar/spelling errors. For instance, in line 121 a verb is missing regarding the starvation period. Sometimes unnecessarily capital letters were used like for instance in line 180 "Ice bath". Please revise also some wrongly used sentences, like for example, line 306 "As the results are shown in table 2...".
2) Check whether all genus and species names are correct. For example, line 425 Hermetia illucens should be in italics.
3) Check the whole manuscript for double spaces or lacking spaces.
Author Response
- During the text it is not entirely clear whether results are expressed based on dry or wat matter. Please check throughout the whole manuscript as it can be a considerable difference.
Response: The results for larval moisture are expressed based on fresh weight.
All other results are expressed based on dry matter.We have clarified whether the results are expressed based on dry or wet matter throughout the manuscript(lines108-115).
- Some abbreviations are not written in full before the first use. This unnecessarily complicates the text for the reader.
Response: Based on your suggestion, we have carefully reviewed all the abbreviations in the text to ensure that each abbreviation is fully spelled out before its first use in the text.
- Please improve the captions of the figures and tables.
Response: According to your suggestion, we have adjusted the captions of all pictures and tables in the manuscript and highlighted them in red in the revised manuscript. At the same time, we have enhanced the size and resolution of Figure 1 and all other graphics.
- Check all citations in the text. Some are wrongly presented such as in line 70.
Response: According to your suggestion, we have carefully checked all references involved in the text and corrected the inaccuracies therein.
- Be consistent in the usage of a space between a number and a unit. Check the consistency in the whole manuscript.
Response: According to your suggestion, we have standardized the spaces between all numbers and units in the manuscript to ensure that a single space is used for separation.
- Check all headings of the manuscript. There are two different ways of presenting used. You have to choose to provide all headings with or without capitals, please revise.
Response: According to your suggestion, we have carefully checked all the titles in the text, and have adjusted the format to use all uppercase letters.
- Check all chemical formulas. Some are wrongly presented such as in line 163 (H2O2).
Response: According to your suggestion, we have carefully checked the expression of all chemical formulas to ensure their accuracy. Thank you for your attention and guidance.
- Explain the difference between sample control group and the control group better. This is not completely clear now.
Response: According to your suggestion, we have made a more detailed and clear description of the differences between the sample control group and the control group in the metabolome analysis section. For details, please refer to lines 388 to 455.
- Please elaborate on the statistical data analysis. You state that an ANOVA is used, without checking the conditions of normality and homoscedasticity. Please have a look at it and make appropriate changes.
Response: According to your suggestion, we elaborated on the statistical data analysis, including the normality and homovariance conditions of the analysis of variance (lines 245-250).
- Check and correct all significant digits in tables and the text.
Response: According to your suggestion, we have standardized the data in the manuscript and adjusted the decimal places of all significant figures to two digits after the decimal point.
- I think there is made a mistake in the caption of figure 1 (two times positive ion mode) and figure 4 (it seems that the instruction of the template or still there).
Response: We are deeply sorry for such a low-level mistake. In the revised manuscript, we have corrected the "positive" in line 374 to "negative", and deleted the template content in the title of Figure 4 (lines 491-492).
- Is there any evidence why the relative concentrations of AA, OA, carbohydrates and vitamins are not consistent? This is not completely clear, please elaborate on this phenomenon (Lines 457-461).
Response: We provide additional explanations for the inconsistency in the relative concentrations of AA, OA, carbohydrates, and vitamins in lines 447-455.
- In lines 476-485 the OPLS-DA is discussed, but it is not completely clear for non-experts. Please elaborate on this to make it more clear for the reader.
Response: According to your suggestion, we elaborated the opls-da analysis on lines 476-485 to make it clearer to non experts.
- Check the whole text for grammar/spelling errors.
Response: We thoroughly checked the grammar and spelling errors in the original and made necessary corrections.
- Check whether all genus and species names are correct. For example, line 425 Hermetia illucenss should be in italics.
Response: According to your suggestion, after carefully reading and checking the full text, all genus and species names in the manuscript are corrected and indicated in italics. In line 396 of the revised manuscript, "Hermetia illucens" has been revised to "Hermetia illucens".
- Check the whole manuscript for double spaces or lacking spaces.
Response: According to your suggestion, after carefully reading and checking the full text, double spaces or missing spaces in the manuscript were corrected.
In addition, some other errors in the original manuscript, such as the journal names of some references have no abbreviations, and there are additional underscores in "DIO", which are corrected after careful reading and checking the full text.
Thank you again for your suggestion. And looking forward to your positive response.
Round 2
Reviewer 2 Report
Comments and Suggestions for Authors
The authors have adeptly addressed both my initial feedback and that of other reviewers. I have no further recommendations to enhance the paper's quality. Thank you.